# Modeling and Evaluation of an Energy-Saving Backpack with Adjustable Stiffness

**DOI:** 10.3390/s25103099

**Published:** 2025-05-14

**Authors:** Jiyuan Wu, Zhiquan Chen, Yinglong Zhang, Xingsong Wang

**Affiliations:** 1School of Mechanical Engineering, Southeast University, Nanjing 211189, China; wujy@seu.edu.cn (J.W.); zhiquanchen@seu.edu.cn (Z.C.); ylzhangfox@foxmail.com (Y.Z.); 2Nanjing Mindray Bio-Medical Electronics Co., Ltd., Nanjing 211111, China

**Keywords:** backpack, human locomotion, energy saving, adjustable stiffness mechanism, adaptability

## Abstract

Backpacks are widely used as an efficient and convenient means for manual load transportation. However, carrying heavy loads for a long time can significantly increase the risk of health issues. In response to the growing demand for relieving muscle fatigue, this paper proposes an energy-saving backpack that can adapt effectively to variable walking speeds and load masses. Inspired by the traditional bamboo pole commonly used for transporting goods, an energy-saving theory based on its mechanical characteristics is proposed. Guided by the theory, the backpack is designed with adjustable stiffness to enhance adaptability across different usage scenarios. Under the experimental conditions of a load of 12 kg and variable walking speeds, the backpack achieves a minimum reduction of 8.6% in the root mean square (RMS) value of gastrocnemius muscle activation. Furthermore, when the load increases from 9 kg to 12 kg, the net metabolic rate is reduced by an average of at least 14.3% compared to conventional backpacks. The experimental results confirm the effectiveness of the proposed backpack under variable conditions, demonstrating the high adaptability and flexibility that the energy-saving backpack provides.

## 1. Introduction

Backpacks are essential tools in daily activities, such as outdoor explorations and occupational tasks. However, prolonged or excessive load carriage places substantial physical strains on the human body, often leading to postural variability [1,2,3], such as increased forward lean of the trunk [4,5], unstable postural sway [6], and variability in stride length and stride time [7,8,9]. These changes in posture and gait lead to uneven force distribution across the human musculoskeletal system, elevating the risk of chronic issues such as lumbar strain, spinal misalignment, and joint pain [1,10,11,12]. Additionally, the increased mechanical stress on muscles and tendons, especially in the shoulders, lower back, and lower limbs, may result in fatigue, soreness, and long-term muscle damage [13,14,15,16,17]. These challenges reveal the urgent need for appropriate ergonomic solutions to alleviate the physical burden when carrying loads while maintaining work efficiency.

To address these issues, researchers have developed different types of energy-saving backpacks. A prominent category includes active dynamic suspended backpacks, which employ mechanical suspensions to actively decouple the load motion from the user gait. These designs aim to reduce vertical oscillations and peak forces on human shoulders [18], leading to a measurable reduction in metabolic energy expenditure [19,20]. For example, He et al. [21] applied a motor to actively control the acceleration motion of the load. The research reduced the vertical acceleration of the backpack by 98.5% to minimize the inertial force generated by the backpack, and the results show a 11% reduction in metabolic power. Park et al. [22] used the active module to adjust the load distribution transferred from the shoulders to the pelvis. However, active systems are often criticized for their added weight and mechanical complexity, which may offset their energy-saving benefits in practical scenarios. Moreover, poor battery life limits the effectiveness of the active backpacks for long-time walking requirements. An alternative approach involves passive suspension systems, which are valued for their simplicity and energy-efficient designs that do not rely on external power sources. Such backpacks often incorporate elastic elements to decouple load motion from human motion to reduce transmission of load-induced forces to the body. For example, Rome et al. [23] introduced a suspended-load backpack using vertical springs to change the load motion during walking, achieving a 23% reduction in metabolic power compared to the typical rigid backpack. Huang et al. tested their passive backpack during walking and running at a constant speed, respectively, and the results exhibited a significant decrease in metabolic rate in both conditions [24]. However, current passive backpack designs face notable limitations. Specifically, some current designs are optimized for a narrow range of gait patterns (e.g., constant-speed walking) and fixed-load conditions [24,25,26], while others introduced motors to build a quasi-passive stiffness adjustment system for specific load weights or walking frequencies [27]. The elastically suspended backpack produced by Foissac et al. [28] not only had no improvement in energy efficiency at low speeds but also significantly increased energy cost at high speeds instead, owing to the large stiffness resulting in the resonant frequency approaching the step frequency. These limitations significantly reduce the energy-saving effects of the backpack in practical, dynamic scenarios involving varying gaits, such as running or walking at variable speeds, and diverse load weights. Therefore, there remains a pressing demand for a more versatile passive energy-saving backpack capable of adapting to both gait variability and load changes.

Although power machinery has largely replaced human and animal labor in modern society, bamboo poles are still used in regions such as East Asia [29], especially in terrains inaccessible by vehicles. However, there are only a few studies on the biomechanics and energetics of bamboo poles. One notable research includes the study by Kram [30], who described a dynamic interaction between the pole and the human shoulder. The results showed that a pole with lower stiffness can reduce the peak force acting on the shoulder, while the metabolic cost was increased and approximately proportional to the load mass. Potwar et al. [31] investigated the influence of different bamboo lengths and shapes on peak shoulder force during walking and running. However, the experiments by Potwar did not measure the metabolic costs of the bamboo poles of different lengths and stiffness recommended. Another research conducted by Castillo [32] compared the metabolic costs of bamboo poles and steel poles and found that the use of bamboo poles reduced the metabolic cost by approximately 5% compared to the use of steel poles. Although there are many contradictions between these studies, it can be seen that the bamboo pole and the carrier gait in the experiment have a certain impact on the metabolic cost of load carriage. Although these studies have explored the biomechanics and energy efficiency of the bamboo pole, the dynamic properties of the bamboo pole have yet to be methodically integrated into modern backpack designs. The large size of the bamboo pole limits its widespread application, while the environmental adaptability needs further exploration.

Given the research status, this study aims to develop a passive energy-saving backpack that adapts to various walking speeds and load masses, enabling users to walk freely while carrying loads. To achieve this, an energy-saving theory based on the dynamic motion of bamboo poles is firstly established to determine the backpack stiffness. Then, a backpack with adjustable stiffness is developed following the energy-saving theory to enhance its adaptability to human locomotion. Lastly, experimental validation is conducted to evaluate the effectiveness of the proposed backpack in load-carrying tasks.

## 2. Development of Energy-Saving Theory

### 2.1. Theoretical Modeling

Previous studies have mainly analyzed the peak force of bamboo poles on human shoulders [30,31] while overlooking the influence of lower limbs in load transportation. Since the human legs provide the mechanical power for propulsion, it is crucial to consider the bamboo pole, the load, and the user as a unified system. This study extends the understanding of bamboo pole mechanics by analyzing the relationship between the dynamic peak forces of the overall system and the energy expenditure of human lower limbs.

The dynamics data of the bamboo pole were captured and measured using a Motion Capture System (Qualisys, Sweden). The data illustrated a clear phase difference between the displacement and acceleration of the center of mass (CoM) of the carrier and that of the load at different walking speeds (Figure 1a,b). As the walking speed increased, the amplitude of movement of the carrier CoM increased, while the load CoM amplitude decreased, with the phase difference between the two approaching 180°. This out-of-phase vertical motion between the carrier and load can be considered a spontaneous internal adjustment in the human–pole–load system, promoting system stability. Then, experiments were conducted using bamboo pole to carry loads, along with control groups that used fixed loads with a high-rigidity pole of the same length. The electromyography (EMG) of gastrocnemius muscles (GAS) and the metabolic rate of the user were measured. As shown in Figure 1d, when using the bamboo pole, the EMG signal of the GAS muscle exhibited lower muscle activation than the control group, and the net metabolic cost was reduced by at least 6.9% compared to the control group.

Based on the observed performance of the bamboo pole, the human–load–pole system is modeled as illustrated in Figure 2. The bamboo pole can be simplified as a flexible cantilever beam, and its self-weight is considered negligible relative to the carried load. During walking, the suspended loads connected by ropes experience horizontal displacement due to inertial forces. However, the rotation angle between the ropes and the coronal plane of the human body is relatively small. Since the primary focus of the energy-saving backpack design is on the vertical motion of the center of mass (CoM) of the load, both the horizontal displacement and rotational angle of the load are neglected in the model for simplification.

Taking the shoulder support as the coordinate origin ($O_{O}$), the kinetic equation of the carrying pole can be obtained as follows:(1)$$m\ddot{y}+c_{eq}\dot{y}+k_{eq}y=k_{eq}y_{0}+c_{eq}\dot{y_{0}}$$
where *m* is the whole mass of the load, $k_{eq}$ is the equivalent spring constant, and $c_{eq}$ is the equivalent damping constant of the compliant bamboo pole. *y* represents the vertical motion of the load, and $y_{0}$ stands for the vertical motion of the human body.

Based on the vertical motion performance of the human CoM, $y_{0}$ can be described as follows:(2)$$y_{0}=Y_{0}sin\left(\omega t\right)$$
where ω represents the walking frequency of the carrier. The equivalent spring constant of the whole bamboo pole can be described as follows:(3)$$k_{eq}=\frac{12EI}{b^{2}\left(3l-b\right)}$$
where *E* denotes the elastic modulus, *I* is the moment of inertia of the bamboo pole, *b* represents the horizontal distance from the load to the center of mass of the human body, and *l* is the half length of the bamboo pole. The movement of the human CoM during walking can be approximated as a sinusoidal curve based on our experiments. Thus, the particular solution of *y* can be expressed as follows:(4)y=Ysin(ωt−ϕ)

The displacement transmissibility of the load and the human body motion can be calculated as follows:(5)$$|\frac{Y}{Y_{0}}|=\sqrt{\frac{1+{\left(2\zeta \gamma \right)}^{2}}{{\left(1-\gamma^{2}\right)}^{2}+{\left(2\zeta \gamma \right)}^{2}}}$$
where the natural frequency of the load is $\omega_{l}=\sqrt{\frac{k_{eq}}{m}}$, the frequency ratio is $\gamma =\frac{\omega }{\omega_{l}}$, and the damping ratio is $\zeta =\frac{c_{eq}}{2m\omega_{l}}$. The phase angle can be derived as(6)$$\phi =arctan[\frac{2\zeta \gamma^{3}}{1+\left(4\zeta^{2}-1\right)\gamma^{2}}]$$

Since the weight of the bamboo pole and the load is applied to the carrier, the muscles need to provide sufficient support to maintain the walking balance and the movement of the CoM of the whole system. The support force $F_{s}$ can be considered as the sum of the gravity of the carrier and load, and the oscillation force generated by the whole system. $F_{s}$ can be described in Equation (Equation 7). The oscillation force is generated by the mass and acceleration of the whole system, as given by Equation (Equation 8).(7)$$\begin{array}{c} F_{s}=Mg+Ma_{sys} \end{array}$$(8)$$\begin{array}{c} F_{osc}=Ma_{sys}=m_{h}a_{h}+ma_{l} \end{array}$$
where *M* is the mass of the whole system, and $M=m_{h}+m$. $m_{h}$ is the human body weight. $a_{h}$ is the acceleration of the human’s CoM, and $a_{l}$ is the acceleration of the load’s CoM.

To simplify the calculation, a mass scaling factor η is introduced, let:(9)$$m=\eta m_{h}$$

The magnitude of the oscillation force $|F_{osc}|$ can be obtained as follows:(10)$$|F_{osc}|=m_{h}Y_{0}\omega^{2}\sqrt{1+2\eta \frac{Y}{Y_{0}}cos\phi +\eta^{2}{\left(\frac{Y}{Y_{0}}\right)}^{2}}$$

This equation can be used as a generalized equation for calculating the magnitude of the oscillation force for a variety of load-carrying walking situations. When using a traditional backpack for load-carrying (fixed-load system), the equation can be simplified to the following:(11)$$|F_{osc-fixed}|=\left(\eta +1\right)m_{h}Y_{0}\omega^{2}$$

### 2.2. Energy-Saving Theory

Mechanical work is an important indicator for evaluating human energy consumption. In the human–pole–load system, the external mechanical work produced by the oscillation force $F_{osc}$ can be expressed as follows:(12)$$W_{osc}=\int F_{osc}v_{CoM}dt$$
where $v_{CoM}$ represents the velocity of the human–pole–load system. This part of the external mechanical work is balanced by the additional energy generated by the carrier.

It is assumed that using the bamboo pole can reduce the energy expenditure of the carrier, and therefore, the external mechanical work generated by the human–pole–load system should be less than that generated by the traditional fixed-load system. An oscillation force scaling factor Q-value is introduced, which represents the ratio of the oscillation force in the human–pole–load system to that in a normal load-fixed system and is described as follows:(13)$$Q=\frac{|F_{osc}|}{|F_{osc-fixed}|}=\frac{\sqrt{1+2\eta \frac{Y}{Y_{0}}cos\phi +\eta^{2}{\left(\frac{Y}{Y_{0}}\right)}^{2}}}{\eta +1}$$

If the scale factor Q-value is between 0 and 1, the oscillation force $F_{osc}$ of the human–pole–load system is smaller than the oscillation force of the fixed-load system $F_{osc-fixed}$, thereby requiring less external mechanical work from the human body. Conversely, when the Q-value exceeds 1, the human–pole–load system generates a greater oscillation force on the carrier, resulting in greater energy consumption.

Based on the experimental data of the bamboo pole, the calculated Q-values for the human–pole–load system range from 0.94 to 0.99, indicating a reduction in the oscillation force amplitude of the system $|F_{osc}|$ by using the bamboo pole. From a kinetic perspective, reducing the peak acceleration of the system lowers the energy consumption of the human body [33]. A decrease in the magnitude of the oscillation force leads to a smaller vertical acceleration fluctuation of the system CoM, which in turn reduces the mechanical energy conversion of the system, thereby lowering energy consumption for the carrier, as reflected by the decrease in oxygen consumption. From a biomechanical perspective, the lower limb muscles contract to provide the necessary support force $F_{s}$ to maintain the CoM movement of the carrier. A lower demand for support force can lead to a decrease in muscle activity. This explains the reduction in root mean square (RMS) values of the gastrocnemius muscle (GAS) when using the bamboo pole.

Therefore, an energy-saving passive control strategy aims to keep the Q-value within the range [0, 1] to achieve a noticeable energy-saving effect.

## 3. Design of Energy-Saving Backpack

In this section, a passive energy-saving backpack with adjustable stiffness is introduced using the aforementioned energy-saving theory. Compared to the motor-driven design, the passive energy-saving backpack offers advantages such as smaller size, lighter weight, and lower cost. Moreover, an effective variable stiffness design can ensure a wider range of adaptability for the backpack.

### 3.1. Parameter Selection

In order to design an energy-saving backpack, it is necessary to determine the stiffness of the backpack. According to the energy-saving theory discussed in the previous section, let(14)0≤Q≤1

In real-world applications, both the carried load mass and walking speed are always changing. Figure 3 illustrates the relationship between the parameter γ and the Q-value under different load masses, assuming fixed backpack stiffness and damping. The shaded region indicates the effective range where the Q-value remains below 1. As walking speed increases (that is, as γ increases), the Q-value initially rises, then undergoes a sharp drop, and subsequently returns to a stable value. Notably, heavier load masses result in a more obvious drop in the Q-value, as well as a lower final stabilized value.

In the backpack design, it is necessary to consider the applicable range of the backpack. Referring to the walking speed of adults during daily activities, the speed range that the backpack can cover is set between 1.25 m/s and 1.9 m/s. Considering the common scenarios in daily life (such as work, travel, etc.), the load mass range that the backpack can cover is set between 5 kg and 12 kg. Therefore, these two parameters can be described as follows:(15)$$\begin{array}{c} 1.25\;m/s\leq v\leq 1.9\;m/s \end{array}$$(16)$$\begin{array}{c} 5\;\mathrm{kg}\leq m\leq 12\;\mathrm{kg} \end{array}$$

Therefore, contour maps of the Q-value influenced by both walking speed (*v*) and backpack load mass (*m*) are plotted. The following figure illustrates the contour variations for different k values. Specifically, two sets of *k* values are studied, including 1000 N/m and 2000 N/m. The shaded areas in Figure 4 represent the combination of walking speed (*v*) and backpack load mass (*m*) where the Q-value is less than 1. It is evident that as the system stiffness ($k_{eq}$) decreases, the contours move towards the lower left, resulting in a larger shaded area. This indicates that a lower system stiffness increases the possibility of achieving a Q-value less than 1, thereby enhancing energy-saving effects.

According to quasi-static equilibrium, the initial static deflection $x_{s}$ caused by the load gravity can be described by Hooke’s law as $mg=k_{eq}x_{s}$. However, a small $k_{eq}$ value would introduce a large initial static deflection $x_{s}$ to the backpack, affecting the human gait. Consequently, the stiffness $k_{eq}$ should be limited as follows:(17)$$\begin{array}{c} k_{eq}\geq \frac{mg}{x_{s}} \end{array}$$

To ensure that the movement range of the backpack does not affect the balance of human walking, referencing the displacement of the human CoM during walking, the static deflection of the backpack is set to be no more than 12 cm.

Figure 5 illustrates the range of stiffness values ($k_{eq}$) for the range of load masses and walking speeds (Equations (15) and (16)). The red surface represents the maximum stiffness values $k_{max}$, which are calculated by Equations (1)–(14), while the blue surface represents the minimum stiffness values $k_{min}$, which are calculated by Equation (Equation 17). The region between these two surfaces indicates the acceptable range of stiffness values. Since no single constant $k_{eq}$ value can achieve energy savings across all combinations of walking speeds *v* and backpack load masses *m*, the energy-saving backpack needs to adopt an adjustable stiffness approach to meet the requirements for achieving energy savings under all combinations of *v* and *m*.

For adjustable stiffness, several key issues need to be addressed. First, users do not maintain a constant walking speed and cannot accurately assess their walking speed during daily walking activities. Therefore, adjusting the stiffness of the backpack in real time based on different walking speeds is inefficient for passive control. Instead, the stiffness of the backpack should ensure energy-saving effects across a wide range of walking speeds, allowing users to walk freely without constraints. Second, the load mass may change as users add or remove items from the backpack. Therefore, changes in the load mass should not affect the energy savings of the backpack. As shown in Figure 5, for a given combination of load mass and walking speed, there exists a valid stiffness interval [$k_{min},k_{max}$] that satisfies the condition of Equation (Equation 14). Furthermore, it is also observed that when the load mass is unchanged, the stiffness intervals corresponding to different walking speeds exhibit substantial intersection. For example, with a 5 kg load, as the walking speed increases from 1.25 m/s to 1.9 m/s, the lower limit of the stiffness interval $k_{min}$ remains the same, while the upper limit of the stiffness interval $k_{max}$ increases. This observation confirms the existence of stiffness values that are valid across the whole range of walking speeds. However, Figure 5 also shows that no single stiffness interval exists that satisfies all load masses, which means that large changes in load mass require stiffness adjustments.

To optimize performance, the load mass range is divided into three intervals: 5 kg ≤m≤ 7 kg, 7 kg ≤m≤ 9 kg, and 9 kg ≤m≤ 12 kg. Then, for each load mass interval, the valid stiffness range that accommodates both the load mass and walking speed ranges is determined as follows:(18)$$k\in \left\{\begin{array}{cc} {}[573.47,\;607.31]\;N/m & m\in [5,\;7]\;\mathrm{kg},\;v\in [1.25,\;1.9]\;m/s\\ {}[736.14,\;840.24]\;N/m & m\in [7,\;9]\;\mathrm{kg},\;v\in [1.25,\;1.9]\;m/s\\ {}[982.08,\;1068.00]\;N/m & m\in [9,\;12]\;\mathrm{kg},\;v\in [1.25,\;1.9]\;m/s \end{array}\right.$$

Based on the results, three specific stiffness values corresponding to each load mass interval are selected: 600 N/m, 800 N/m, and 1000 N/m, respectively. Within each load mass range, there is no need to adjust the stiffness when adding or removing the load mass. The stiffness needs to be adjusted only when the load mass changes beyond the range interval. At the same time, during the load transportation process, the user can freely change the walking speed without considering the backpack stiffness issue.

### 3.2. Backpack Design

Figure 6a illustrates that the energy-saving backpack consists of a moving system and a back system. The back system is connected to the shoulder straps and fixed to the body. The load is connected to the moving system. A leaf spring structure similar to the bamboo pole is applied to the back system. The center of the leaf spring is fixed to a compression spring. The lower end of the compression spring is coupled to the back plate. Two sliders are attached to the leaf spring by a sliding connection. The rods form a deformable rhombus mechanism, in which the lower end connects to the roller screw mechanism. The connectors on the left and right sides of the rhombus mechanism are coupled to the sliders. The linear guide and the pulley can limit the moving plate to move in the vertical direction.

In this design, the stiffness is adjusted by modifying the effective length of the cantilever beam of the leaf spring. The rhombic mechanism transfers the load weight to the spring by the sliders. Users can rotate a shaft to drive a roller screw, thus reshaping the rhombic mechanism and adjusting the slider position on the leaf spring. Once the shaft rotation stops, the roller screw ensures the rhombic mechanism remains self-locking in its current state, preventing further movement of the slider. Figure 6b shows the stiffness adjustment from 600 N/m to 1000 N/m and the corresponding positions of the slider, respectively. When the slider moves away from the center of the leaf spring, the oscillation length of the cantilever beam increases, resulting in a decrease in system stiffness. Figure 6c illustrates the relationship between the backpack stiffness and the slider position relative to the center of the leaf spring. Although the system supports continuous stiffness adjustment, this process is not necessary. In the case of daily walking at unsteady speeds with a load mass within the interval of [5 kg, 7 kg], using a stiffness of 600 N/m can ensure energy-saving effects. When the load mass is still within this load range after increasing or decreasing, there is no need to adjust the stiffness. Only when the load mass increases to more than 7 kg does the backpack need to adjust the stiffness to 800 N/m.

## 4. Experiments and Results

### 4.1. Experimental Setup

To validate the effectiveness of the energy-saving backpack on real users, experimental tests involving six healthy participants (age 25; weight 70 ± 5 kg) were conducted. All studies involving human subjects were conducted following the principles of the Declaration of Helsinki. Consent was obtained from the subjects after the details and possible consequences of the study were explained.

Each subject walked on a treadmill under two conditions: coupled backpack—the subject walked with a load and an energy-saving backpack, but the stiffness adjustment system was turned off and the backpack acted like an ordinary backpack; and flexible backpack—the subject walked with the full energy-saving backpack. Before testing, each subject completed a 5 min session of adaptive load-carrying walking under the two conditions. The backpack weighed 1.8 kg, which was not included in the load weight in the experiments. The order of the experiments under the two conditions was randomized for each subject. The duration of each trial was 7 min. Data were collected for the last 5 min. A ten-minute break was taken between the two trials. Figure 7 shows the experimental test scenario.

The gastrocnemius muscle plays an important role in generating forward propulsion and can be modulated in response to changes in external loads [34]. Therefore, the gastrocnemius muscle was selected as the object of observation of energy consumption. Muscle activities were recorded at 1000 Hz using a wireless electromyography (EMG) system (Delsys Trigno, Natick, MA, USA). Raw EMG data underwent a fourth-order band-pass Butterworth filter (15 to 280 Hz) to eliminate noise signals and biological artifacts. The signals were full-wave-rectified, followed by low-pass filtering (fourth-order Butterworth, 7 Hz) to create a linear envelope. Processed EMG time-series signals from 10 complete strides in each condition were selected. To quantitatively assess changes in muscle activities, root mean square (RMS) was calculated for the processed EMG signals. The RMS values for the flexible backpack were compared with those for the coupled backpack to evaluate changes in muscle activities.

The metabolic rate of each subject was assessed using a portable wearable metabolic system (Cosmed, Albano Laziale, Italy) by measuring $O_{2}$ consumption and $CO_{2}$ production. Brockway’s standard equations were employed to convert these values into metabolic rate [35]. Post hoc paired *t*-tests were conducted to identify significant changes in metabolic cost, with a significance level set at 0.05 for all analyses.

Before the formal experiments, a pilot study was conducted to evaluate the effectiveness of the experimental sample size. An a priori power analysis was performed using G*Power (version 3.1.9.7) [36] based on the metabolic rate data from six participants under the two backpack conditions with a load mass of 12 kg and walking speed of 1.25 m/s. The paired *t*-test was selected for comparison. The average metabolic rate reduction with the flexible backpack was 11.9% ± 3.1% (mean ± SD), corresponding to a large effect size (Cohen’s d = 3.83). Using a significance level of 0.05 and a desired power of 0.8, the analysis indicated that a minimum of four participants would be sufficient. Therefore, the sample size in the formal experiments is statistically adequate to support the observed effects.

### 4.2. Results

Figure 8a displays that when carrying the same load (12 kg), as the walking speed increases, the CoM acceleration shows a significant decreasing trend. Additionally, the phase difference of the motion at all three speeds is close to 180°. At the same speed (1.38 m/s), under three different backpack stiffness values, with loads in the corresponding load range, the flexible backpack can achieve out-of-phase motion between the load and the human body (Figure 8b).

For the flexible backpack, the oscillation force exhibits a significant redistribution, as shown in Figure 9. For ease of comparison, the oscillation force is standardized, i.e., the ratio of the oscillation force to the body weight (BW). During a gait cycle, the oscillation force of using the flexible backpack shows an average decrease of 38.2% in peak push-off and an average increase of 58.7% in midstance compared to the coupled backpack. Additionally, the positive mechanical work rate is reduced by 22.6% on average compared to the coupled backpack.

The performance of GAS is compared when using the flexible backpack and the coupled backpack at different walking speeds. The normalized data from six healthy subjects, along with the averages, are presented in Figure 10. It is evident that at the three different speeds, the normalized RMS data of the GAS for the flexible backpack decrease by 11.7%, 8.6%, and 10.1%, respectively.

The net metabolic rate performance of the flexible backpack is better than that of the coupled backpack. As shown in Figure 11a, for both the coupled backpack and the flexible backpack, the net metabolic rate significantly increases with speed. However, the net metabolic rate of the flexible backpack is consistently lower than that of the coupled backpack at all speeds tested. At a walking speed of 1.25 m/s, the flexible backpack reduces the metabolic rate by 12.1%, and as the walking speed increases, the net metabolic rate can be reduced by more than 17.4%.

The flexible backpack maintains a lower net metabolic rate under stiffness adjustments and load changes. Figure 11b shows that after adjusting the stiffness from 800 N/m to 1000 N/m, the metabolic performance of the flexible backpack carrying a constant load (9 kg) remains stable, with only a 2.7% variation, indicating minimal metabolic fluctuation due to stiffness adjustments. In the comparison of metabolic rates between different loads at the same stiffness (1000 N/m), the net metabolic rate increases significantly with increasing load mass. However, the flexible backpack carrying a 9 kg load shows an average reduction of at least 14.3% (p<0.05) in metabolic rate compared to the coupled backpack, while the metabolic rate of the flexible backpack carrying a 12 kg load further increases the reduction rate by 4.5%.

## 5. Discussions

### 5.1. Stiffness Adjustment

The adjustable stiffness mechanism proposed in this study addresses limitations found in previous suspension backpack designs. Traditional passive backpacks typically rely on fixed spring-based systems to decouple the motion between the load and the user, which can offer certain energy-saving benefits but are constrained by their narrow operational range of walking speeds and load weights. This limitation arises from the intrinsic structural properties of springs, such as coil number and diameter, which determine their stiffness. Although some studies have integrated electronic control systems to dynamically adjust the spring stiffness by changing the number of coils that can be compressed or stretched, this approach significantly increases the net weight of the backpack due to the addition of motors and control systems [27].

In contrast, this energy-saving backpack is designed with a compact and lightweight adjustable stiffness mechanism based on the working principle of a bamboo pole. Specifically, the ball screw system at the bottom of the backpack enables the user to manually change the position of the contact point between the load and the cantilever beam, thereby effectively changing the stiffness of the leaf spring and the natural frequency of the system. This mechanical stiffness adjustment greatly reduces the weight of the backpack. At the same time, the screw in the ball screw mechanism can keep the ball slider at any position and self-lock to maintain the current state of the rhombus mechanism.

Unlike active suspension backpacks that aim to optimize energy savings through continuous dynamic adjustments in real time [21,22,37], the proposed adjustable stiffness mechanism in the energy-saving backpack adopts a different concept of stiffness adjustments. By selecting three discrete stiffness values to perform effectively across a wide range of walking speeds and load weights, the design prioritizes energy-saving coverage rather than peak efficiency. This approach is particularly advantageous in the real world, where walking speeds often fluctuate unpredictably, challenging the responsiveness of active control systems. Therefore, it is important to ensure that users can freely change their walking speed during load transportation. In the real world, the mass of the backpack load will increase or decrease more or less during transportation. Thus, it is also very important to ensure that the backpack remains in an energy-saving state after the load mass increases or decreases. This paper computationally identifies three key stiffness values including 600 N/m, 800 N/m, and 1000 N/m, which provide energy savings over a range of common walking speeds and loads. While the design mechanism also allows for continuous adjustment of stiffness within a defined range, these three discrete values are sufficient to cover typical use cases.

Through the experimental results (Figure 11), the energy-saving effect and coverage of the energy-saving backpack at different walking speeds and loads can be observed. When carrying a load of 12 kg and using a backpack stiffness of 1000 N/m, the metabolic rates at different walking speeds are significantly reduced, and the energy-saving effect is more obvious when the walking speed is increased from slow speed to fast speed. When the backpack load mass increases from 9 kg to 12 kg during load transportation, the energy-saving effect is also more obvious. Comparing the curves in Figure 3, it can be seen that faster walking speeds and heavier loads make the Q-value smaller and away from the critical value of 1. The experiment also proves that when the stiffness is adjusted, for example, from 800 N/m to 1000 N/m, the energy-saving effect of the backpack is slightly reduced, but it will not have a significant impact on the user experience.

### 5.2. Dynamic Oscillation Force

The proposed energy-saving theory of the backpack focuses on reducing the dynamic oscillation force transmitted to the lower limbs, offering a departure from existing studies that primarily target shoulder peak force reduction [30,31]. While minimizing shoulder peak force through decoupling the vertical motion of the backpack from the ground can reduce discomfort, the actual weight of the load is still inevitably carried by the user. Consequently, the dynamic forces acting on the lower limbs during walking play a critical role in energy expenditure.

According to the inverted pendulum walking model theory, no mechanical work is required during the single-support phase, as the stance leg functions like an inverted pendulum. The only energetic cost occurs in step-to-step transitions [38], primarily in the impulsive push-off and heel-strike collision phases, where mechanical work is needed to redirect the human CoM motion. During this transition phase, the vertical acceleration of the human body reaches a positive peak (Figure 8), directed upward, while the energy-saving backpack with out-of-phase movement exhibits a negative peak value in vertical acceleration, directed downward. This out-of-phase interaction effectively lowers the overall vertical acceleration amplitude of the whole human–pole–load system, thereby reducing the dynamic load on the lower limbs during the push-off phase. According to Equation (Equation 8), the dynamic oscillation force of the human–pole–load system reaches a minimum. In the study by He [21], the vertical acceleration of the backpack was minimized to 0 to reduce inertial forces. However, this approach neglects the directional relationship between human and backpack accelerations. As a result, during the step-to-step transitions, the vertical acceleration amplitude and the oscillation force of the entire human–pole–load system remain higher than those achieved with the out-of-phase strategy proposed in this study. As shown in Figure 9a, the oscillation force peak during the push-off phase (50–60% gait cycle) was reduced by 38.2%. Part of the original oscillation force is redistributed to the single-support phase where no mechanical work is consumed. An average reduction of 22.6% in positive mechanical work rate (Figure 9d) clearly demonstrates that the energy-saving backpack can effectively reduce energy costs during step-to-step transitions, compared to the coupled backpack.

Muscle activity analysis further supports the theory. The gastrocnemius muscle, a primary contributor during the push-off phase, exhibits reduced RMS values under varying speeds (Figure 10). This reduction indicates decreased muscle activation and effort, highlighting the backpack’s effectiveness in alleviating mechanical and physiological strain on the lower limbs.

Although the backpack design mainly targets the dynamics of the lower limbs, it also takes into account the effect on shoulder load. Feedback from users and experiments showed a cyclic pattern of shoulder pressure, with increased forces occurring in the single-support phase and lower forces in the double-support phase. This redistribution of load pressure makes forward movement during step-to-step transitions easier, aiding in carrying loads more easily.

## 6. Conclusions

This study, inspired by the traditional application of compliant bamboo poles for load transportation, proposes a Q-value energy-saving theory for load carriage design. Based on the mechanical modeling of bamboo poles, a stiffness adjustment structure is applied for the energy-saving backpack, allowing users to carry loads with greater autonomy, adapting to various walking speeds and load changes. Experiments at different walking speeds and loads demonstrated energy-saving benefits. The RMS of the gastrocnemius muscle (GAS)’s EMG activity decreased by at least 8.6% across all walking speeds. Additionally, metabolic tests showed that the energy-saving backpack reduced the net metabolic rate by at least 12.1% compared to the traditional backpack.

In future research, efforts will focus on further enhancing the performance of the energy-saving backpack and expanding its load capacity range to meet the demands of varying levels of labor intensity.

## Figures and Tables

**Figure 1 sensors-25-03099-f001:**
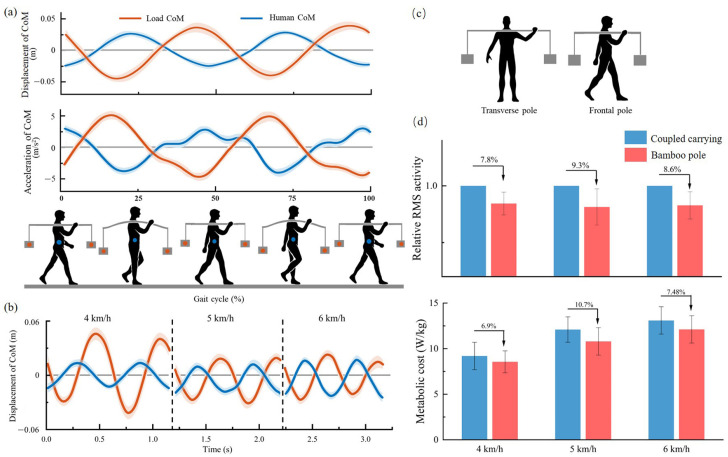
Performance data of the bamboo pole carrying a 15 kg load. (**a**) Variations in displacement and acceleration of the load and human CoM at 5 km/h. (**b**) Variations in displacement of the load and human CoM at different speeds. (**c**) Two typical postures when the pole is used. (**d**) Comparisons in GAS muscle activity and metabolic rate at different speeds.

**Figure 2 sensors-25-03099-f002:**
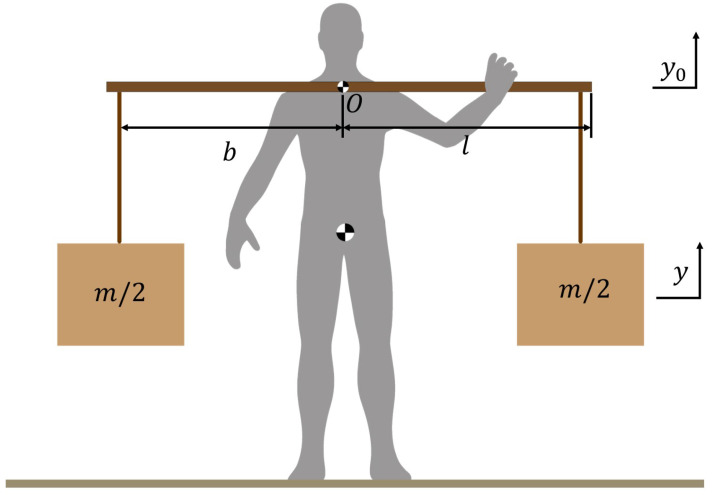
The human–pole–load system. Symmetrical construction of the two ends of the carrying pole.

**Figure 3 sensors-25-03099-f003:**
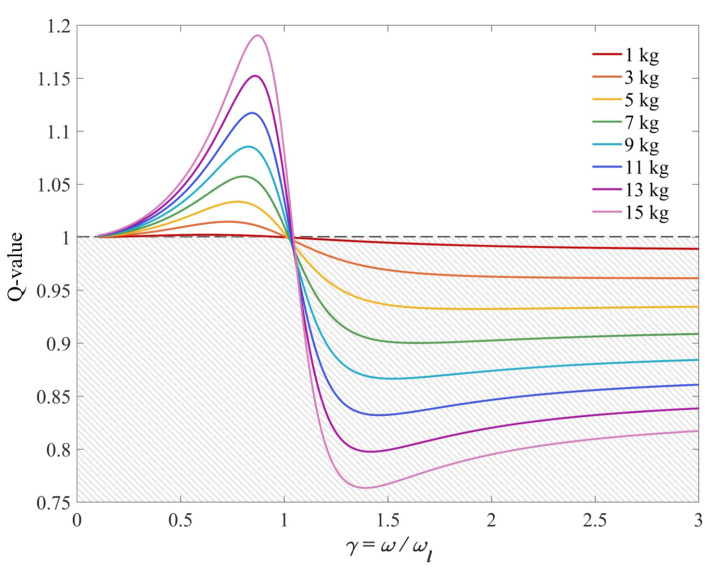
The relationship between Q-value and frequency ratio γ under different load masses. The shaded area represents the effective range of the Q-value.

**Figure 4 sensors-25-03099-f004:**
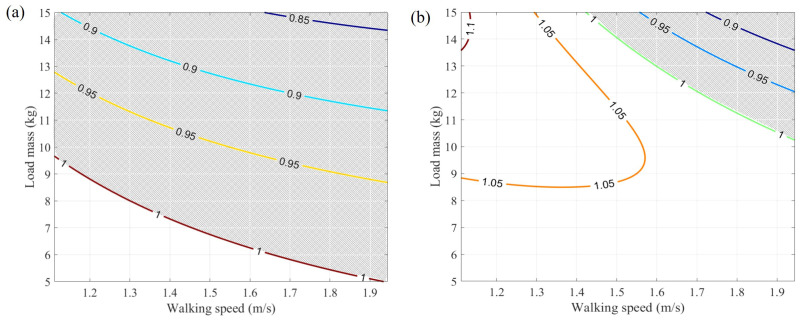
Q-values under different walking speeds and loads. (**a**) Stiffness $k_{eq}=1000$ N/m. (**b**) Stiffness $k_{eq}=2000$ N/m. The shaded areas represent the combinations of speed and load where Q≤1.

**Figure 5 sensors-25-03099-f005:**
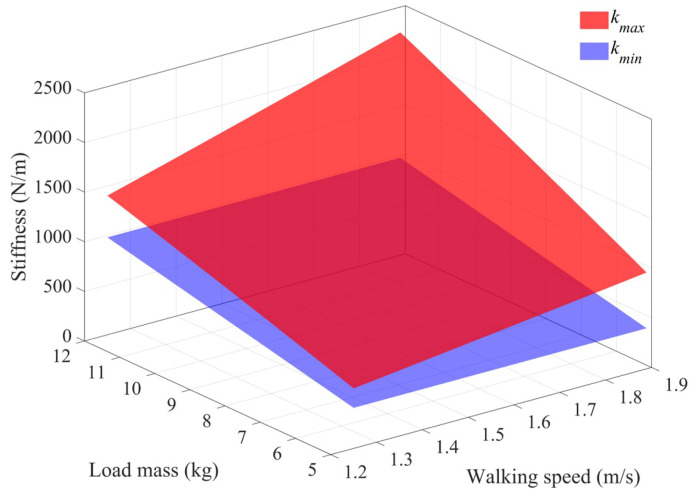
The range of stiffness ($k_{eq}$) values under the walking speeds range of [1.25, 1.9] m/s and load masses range of [5, 12] kg. The red surface represents the upper limit of the stiffness value $k_{max}$, while the blue surface represents the lower limit of the stiffness value $k_{min}$.

**Figure 6 sensors-25-03099-f006:**
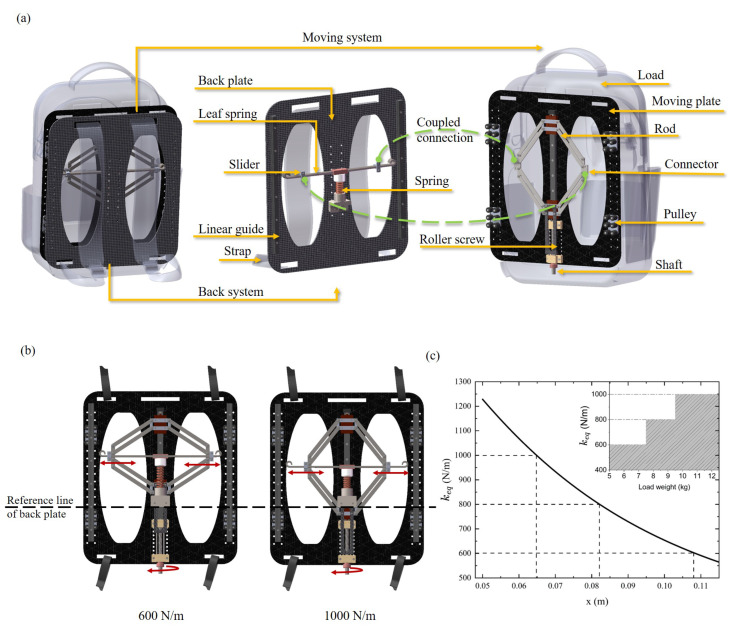
Structure and mechanism of the energy-saving backpack. (**a**) Schematic of the backpack including the moving plate structure and back plate structure. (**b**) Stiffness adjustment from 600 N/m to 1000 N/m. (**c**) The relationships between the distance (*x*) of the load application point relative to the middle and the stiffness change, as well as between the load weight and the chosen stiffness.

**Figure 7 sensors-25-03099-f007:**
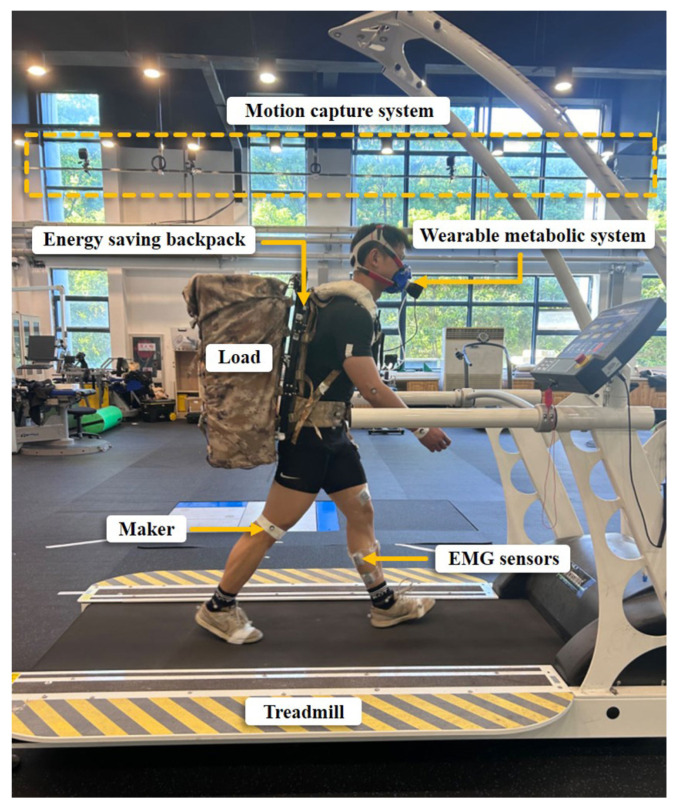
Experimental platform and conditions.

**Figure 8 sensors-25-03099-f008:**
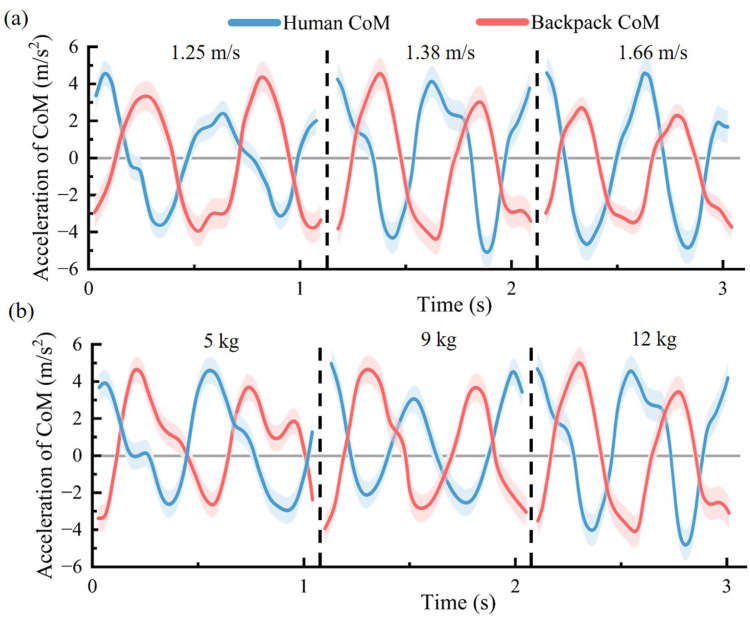
The CoM acceleration performance of human and load. (**a**) Variations at three different walking speeds with the same 12 kg load. (**b**) Variations at three different loads at the same walking speed (1.38 m/s).

**Figure 9 sensors-25-03099-f009:**
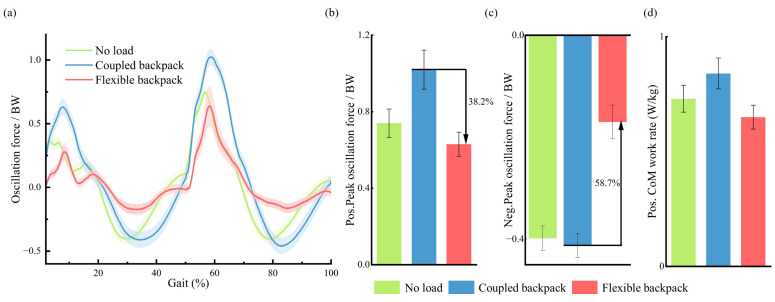
The energetics performance of the flexible backpack. (**a**) Variations in the ratio between $|F_{osc}|$ and BW during a gait cycle. (**b**) Comparisons in the ratio between positive peak $|F_{osc}|$ and BW. (**c**) Comparisons in the ratio between negative peak $|F_{osc}|$ and BW. (**d**) Comparisons in positive CoM work rate.

**Figure 10 sensors-25-03099-f010:**
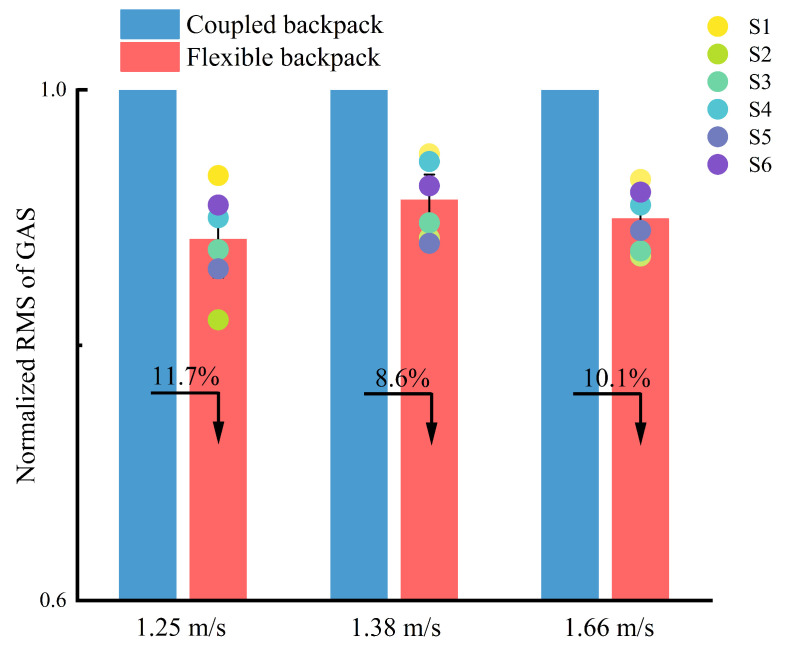
The EMG performance of the gastrocnemius muscle (GAS) when using the flexible backpack and the coupled backpack at different walking speeds. S1 to S6 represent the results from six subjects.

**Figure 11 sensors-25-03099-f011:**
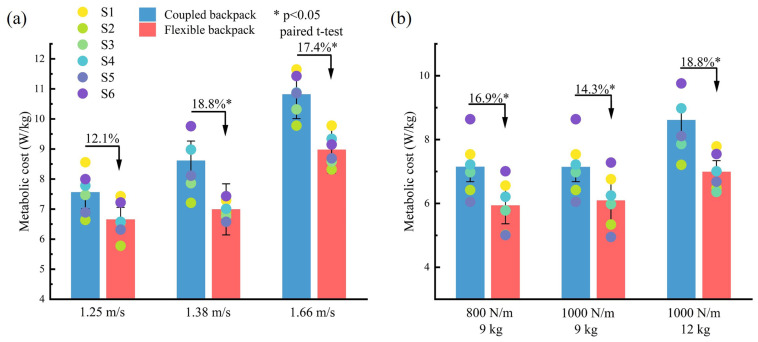
The net metabolic rate performance. (**a**) Same load (12 kg) with different speeds. (**b**) Same speeds with different loads. S1 to S6 represent the results from six individuals. Paired *t*-test, p<0.05.

## Data Availability

The original contributions presented in this study are included in the article. Further inquiries can be directed to the corresponding author.

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
