# Peer review of "Modeling and Evaluation of an Energy-Saving Backpack with Adjustable Stiffness"

_sensors, 2025, doi:10.3390/s25103099_

Round 1

Reviewer 1 Report

Comments and Suggestions for Authors

Based on the situation of traditional flexible bamboo pole transportation, this study designed a backpack with variable stiffness mechanism. On the one hand, this study first expanded the traditional load system for the shoulder, and incorporated the lower limbs into the system for overall discussion. Then, the relationship between variables including bamboo stiffness was analyzed, and the oscillating force and energy consumption during the movement was mathematically modeled, which provides theoretical support for subsequent mechanical design. In addition, a comparative experiment with a backpack without stiffness adjustment was conducted to verify the effect of reduction of energy consumption compared with the ordinary backpack. In total, this manuscript is well written and data well organized. However, some questions need to be addressed before it can be accepted.
1. The article describes a "backpack with a variable stiffness mechanism" in the introduction but frequently uses terms like "backpack with tunable stiffness" or "the stiffness of the backpack" elsewhere. While understandable, these phrases may cause ambiguity or lack rigor. It is recommended to revise such terminology for consistency.  
2. Inappropriate figure grouping leads to excessive distance between figures and their citations (e.g., Fig. 1d spans two pages and another figure: Figs. 10-11). Some figures lack clear annotations (e.g., Fig. 3 omits labels for critical data points; Fig. 9b-d lacks color legends). Please reorganize the figure placement to enhance the annotations in details.  
3. The claim that "lower stiffness introduces larger initial deflection xs" lacks data support. Equation 17, which defines stiffness range requirements, lacks derivation, and xs is not previously defined. Supplementary relevant data or formula derivations need to be provided.  
4. The origin of stiffness upper or lower limits in Fig. 5 is unclear. While Equations 15-16 only mention speed and mass ranges, which do not directly define stiffness. Please clarify the related description.  
5. The assertion of "unnecessity for continuous stiffness adjustment" lacks empirical support. No justification is provided for selecting three discrete stiffness values. Please provided some data to demonstrate the efficacy of chosen stiffness ranges and explain why continuous adjustment is unnecessary.

Reviewer 2 Report

Comments and Suggestions for Authors

This work presents an interesting concept through the development of an energy-harvesting backpack. However, several important issues should be addressed to enhance the rigor and clarity of the manuscript:

  1. Figure 3: The indicators presented in this figure lack sufficient explanation. The authors are encouraged to elaborate on the meaning and relevance of each indicator. Additionally, the design and quality of Figure 3 should be improved to enhance readability.

  2. Figure 5: The range of load mass used in the experiments should be clearly highlighted in the figure or caption, as it is a key variable influencing the outcome and interpretation of results.

  3. Sample Size Justification: While the current study involves six participants, the authors should calculate and report the minimum required sample size using an appropriate statistical power analysis formula. This would support the validity and generalizability of the findings.

  4. Comparison with Related Work: A similar study titled "A Backpack Minimizing the Vertical Acceleration of the Load Improves the Economy of Human Walking" published in IEEE should be discussed in more detail. The authors should clearly differentiate their approach and results from this work, as well as compare their findings with other state-of-the-art energy-harvesting backpack systems in the literature.

Comments on the Quality of English Language

The English could be improved to more clearly express the research.

Round 2

Reviewer 2 Report

Comments and Suggestions for Authors

Authors revised the manuscript accordingly